## REVIEW ARTICLE

# Transcriptome-wide association studies: recent advances in methods, applications and available databases

Jialin Mai[1,2,3,4], Mingming Lu[1,2,3,4], Qianwen Gao[1,2,3], Jingyao Zeng [1,2✉] &
Jingfa Xiao [1,2,3✉]

Genome-wide association study has identified fruitful variants impacting heritable traits. Nevertheless, identifying critical genes underlying those significant variants has been a great task. Transcriptome-wide association study (TWAS) is an instrumental post-analysis to detect significant gene-trait associations focusing on modeling transcription-level regulations, which has made numerous progresses in recent years. Leveraging from expression quantitative loci (eQTL) regulation information, TWAS has advantages in detecting functioning genes regulated by disease-associated variants, thus providing insight into mechanisms of diseases and other phenotypes. Considering its vast potential, this review article comprehensively summarizes TWAS, including the methodology, applications and available resources.

Genome-wide association study (GWAS) has detected significant genetic variants associated with a wide diversity of complex traits, including qualitative traits (e.g., cancer) and quantitative traits (e.g., body height). Beyond GWAS, there are many strategies for gene-set analysis to locate important genes, such as MAGMA[1], FLAG[2,] etc. In recent years, transcriptome-wide association study (TWAS) was proposed and gained a broad range of applications. TWAS is a gene-prioritization approach that detects trait-associated genes regulated by significant variants, primarily single nucleotide polymorphism (SNP) identified from GWAS. Briefly, TWAS first trains a genetic regulation model of genetic components and gene expression from a small available reference panel. These models with regulatory weights are used to impute gene expression for individuals of larger GWAS cohorts. Finally, the associations between predictive gene expression and traits are calculated to determine the regulatory relationship between genes and traits. TWAS has several advantages: It owns (a) Higher gene-based interpretability than GWAS. Although GWAS has identified numerous significant variants, most are in non-coding regions and are hard to interpret. As genes are functioning and more explainable units than variants, the gene-trait associations detected by TWAS are more sensible for explaining the genetic mechanism of phenotypes. (b) Lower computing complexity and experimental cost. Compared to genome loci-based analysis, TWAS conducts association tests only for genes significantly regulated by genetic variations, thus relieving the multiple testing burden[3]. Besides, as it narrows the range of candidate genes, this method can save time and labor costs for following experiments. (c) Tissue specificity. Most disease investigations are based on specific pathological tissues. By targeting relevant and functional tissues, TWAS detects trait-associated gene panels to better reveal the underlying pathological mechanisms for diverse

[1] National Genomics Data Center, Beijing Institute of Genomics, Chinese Academy of Sciences and China National Center for Bioinformation, Beijing 100101, China. [2] CAS Key Laboratory of Genome Sciences and Information, Beijing Institute of Genomics, Chinese Academy of Sciences and China National Center for Bioinformation, Beijing 100101, China. [3] University of Chinese Academy of Sciences, Beijing 100049, China. [4]These authors contributed equally: Jialin Mai, Mingming Lu. ✉email: zengjy@big.ac.cn; xiaojingfa@big.ac.cn

diseases in a tissue-specific manner[4]. (d) Higher statistical power than other gene-based analyses. Facing limited samples with expression data due to finite bio-samples and the high cost of RNA sequencing, TWAS highlights imputing gene expression of large-scale GWAS samples and thus significantly improves the statistical ability of association testing. (e) Leverage genetic regulation information. TWAS builds an expression imputation model from expression quantitative loci (eQTL) data and thus identifies genetically-regulated genes associated with traits, even if they are far from the variants.

TWAS was first introduced by Gamazon et al. in 2015[5] and has snowballed with much attention in recent years (Supplementary Fig. 1a). Researchers have developed multiple computational methods to refine the performance and efficiency of TWAS, and a previous review summarized some typical models until 2020[6]. Current TWAS researches have covered various traits, including cancers, complex diseases such as neurological disorders[7–9], autoimmune diseases[10], and physiological characteristics such as body mass index[11]. However, a more systematic and specialized introduction to TWAS covering its latest method improvements, applications, and available data resources still needs to be addressed.

In this review, we first summarize the principal workflow of TWAS and introduce its developing models, including the fundamental penalized regression-based model and different extensions. We also provide a general collection of eQTL data sources in addition to the widely known GTEx database. Next, we review the practical applications of TWAS toward complex polygenic diseases and physiological phenotypes. Then we summarize available database resources storing statistical results and analysis tools for data integration. Finally, we provide a practical guide for choosing TWAS methods oriented by research aims and public data and then discuss possible future improvements of TWAS based on current limitations and potentials.

## Workflow and updating models of TWAS

TWAS conducts association tests for each gene-trait pair leveraging analysis from eQTL effects and GWAS associations. Based on the idea of TWAS, the basic workflow can be divided into three steps, as illustrated in Fig. 1. It is noted that different models are contained in various stages and should be distinguished carefully by their purposes.

(1) Training stage: this step aims to estimate regulatory effect sizes of multiple SNPs on the gene expression level by fitting a multivariate SNP ~gene expression model from a small reference panel with both genotype and expression data. To our knowledge, many eQTL databases and specific data sets are utilized to build the model (summarized in Table 1). Considering the accession requirement of many data sets, Gamazon et al.[5]. and Gusev et al.[12]. have proposed commonly-used models in their websites, respectively (PredictDB, https://predictdb.org; FUSION, http://gusevlab.org/projects/fusion/). To build an expression prediction model, SUMMIT[13] was developed to utilize eQTL summary statistics and LD information from reference genomes.

(2) Imputation stage: this step aims to obtain the predicted gene expression level of GWAS individuals. The trained prediction model takes large-scale genotype data from GWAS as inputs and outputs the calculated expression level of each gene. Imputation of expression might be combined into the association with GWAS summary statistics data, which will be discussed later. Generally, a reference panel of the same ancestry with GWAS samples is recommended to avoid disturbing unknown linkage disequilibrium (LD) profiles among diverse populations. Recently, multi-ancestry TWAS methods integrating eQTL data from multiple ethnicities have been developed[14,15], enabling cross-ethnic reference with a larger sample size.

(3) Association stage: this step implements hypothesis tests between predicted gene expression and the target trait with different statistic association models. Finally, it pinpoints significant trait-associated genes with their effect sizes calculated. To avoid false positive results induced by multiple testing, statistical corrections are applied to adjust p-values, including Bonferroni correction, Benjamini-Hochberg correction, etc.

Since the first attempt in 2015, different models have been developed to fit available data and enhance TWAS accuracy. The timeline of TWAS methods developments with trait instances is shown in Fig. 2.

**Models used in the training stage**. Different prediction models are based on various assumptions of the estimated SNP weights in SNP ~expression relationships.

At first, considering tissue-specific transcription regulation, the expression prediction is performed in each tissue separately. The pioneer of TWAS, PrediXcan[5], has applied a linear regression model to fit the relationship between multiple cis-SNPs (within 1 Mb around the transcription/end site of the gene) and the target gene expression level. With a reference panel of n individuals, for a given gene g, the relationship of its expression level and corresponding multiple SNP variants can be formulated as:

$$Eg = \mu + X\beta + \varepsilon \tag{1}$$

In Eq. (1), $E_g$ is a n vector that denotes the expression level after correction for confounders (such as age, sex, genotype principle components (PCs) and so on), X is a $n*p$ genotype matrix with p-vector SNPs (coded as 0/1/2 or genotype dosages) on the same set of n individuals. β is a p vector of SNP weights which denotes the corresponding eQTL effect sizes for g. ε denotes the error term and μ denotes the intercept term, which can be dropped after $E_g$ and X are centered at zero. As the number of samples is often less than that of variables (i.e., SNPs), they applied penalized regression models for filtering important SNPs and avoiding overfitting, including lasso (least absolute shrinkage and selection operator) and elastic net. With PrediXcan, the result showed that elastic net and lasso are more accurate and robust in predicting expression levels than simple polygenic models[5]. In particular, the elastic net modeling estimates β with a linear combination of lasso (L1) and ridge (L2) penalties by:

$$\hat{\beta} = \underset{\beta}{\text{argmin}} \left[ ||Eg - X\beta||_2^2 + \lambda \left( \alpha||\beta||_1 + \frac{1}{2}(1-\alpha)||\beta||_2^2 \right) \right] \tag{2}$$

In Eq. (2), $|| \cdot ||_1$ denotes the L1 norm, $|| \cdot ||_2$ represents the L2 norm. α indicates the proportion of L1 penalty, which is usually set as 0.5 (as in PrediXcan). λ denotes the penalty parameter, which is estimated by cross-validation[5]. Afterward, the lasso and elastic net models laid the basic form for predicting gene expression levels, extended in many later TWAS models. Another widely-used software, FUSION, adopts the Bayesian sparse linear mixed model (BSLMM)[12], which is a hybrid of the Bayesian variant selection model (a sparse regression model) and a linear mixed model. The BSLMM model was early used in polygenic modeling for polygenic risk scoring and phenotype prediction[16], then used to estimate the distribution of genetic effect size (β) on a given gene locus in FUSION. In TWAS, the two models hold diametrically opposite assumptions about the distribution of β. The linear mixed model assumes that β is normally distributed. In contrast, the Bayesian variant selection model assumes a sparse distribution of β, which means a small proportion of cis-SNPs have important effects on gene expression. Consequently, the combined model takes advantage of both assumptions and can

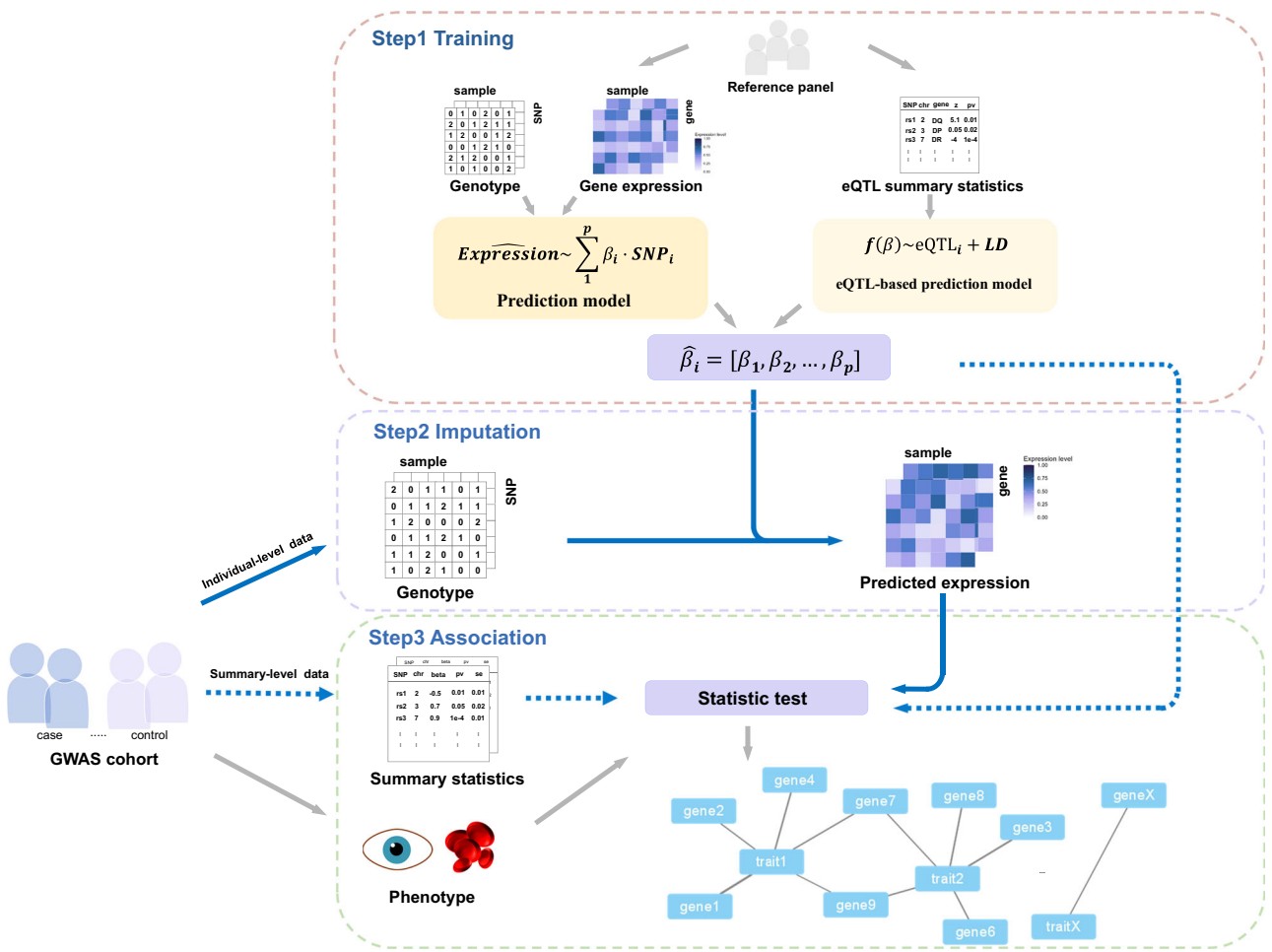

**Fig. 1 Schematic workflow of TWAS analysis.** The first step is to train expression predictive models by inputting either genotype data and corresponding expression data of reference panel, or eQTL summary statistics with specific TWAS methods. Next, for individual-level GWAS data, the second step imputes the expression data of GWAS individuals using the fitted predictive models. The third step analyzes the association between each phenotype-imputed gene expression pair (as the blue solid line shows). The predicted SNP weight vector is combined to calculate association statistics (as the blue dashed line shows) for GWAS summary statistics data.

adapt flexibly to real eQTL effect size distribution when the real genetic mechanism is unknown.

Despite their popularity, PrediXcan and FUSION utilize *cis*-SNPs in their linear prediction models without considering more complex genetic structures. Advanced models attempted to cover more effects of genetic variants to reach higher predictive accuracy. For example, random forest is an ensemble tree-based nonparametric model used to predict the genetic effects of SNPs[17]. In TWAS, the random forest regression model enables a pre-selection of significant SNPs by feature importance measurement with reduced computational complexity and generalization error to train better expression prediction models[18]. Additionally, TIGAR[19] used a non-parametric Dirichlet process regression (DPR) model to capture more genetic effects and achieve more robust performance. The Latent Dirichlet process introduces an unknown distribution on the variance parameter of SNP effect size ($\beta$) and estimates $\beta$ based on inputted data rather than parametric priors. In the paper, the model comparison showed that the non-parametric model had more substantial TWAS power than PrediXcan.

In addition, integrating transcription regulation data is an efficient way to help demonstrate SNP effects and improve prediction models. For example, epigenetic regulation has been found to affect gene expression, so using relevant data can be considered in prediction models. For instance, Epigenetic

element-based TWAS was developed in 2021[20]. It firstly divides all *cis*-SNPs into multiple SNP sets according to their eQTL effects and epigenetic annotations. These epigenetic annotations include chromatin state, transcription factor binding site, and DNase hypersensitive locus. Then lasso and elastic net models are built with different SNP sets as conventional TWAS. Furthermore, considering a more complex regulatory architecture, a Multi-Omic Strategies TWAS (MOSTWAS)[21] approach was proposed. Additional prediction models of regulatory elements embedding these SNPs are first trained to model a given gene and its distal SNPs. The imputed values of regulatory genes are incorporated into the final prediction model. Only chromatin status, transcription factor binding sites, microRNA effects, and CpG methylation sites are currently included, and regulatory elements can be further extended. Consequently, MOSTWAS is featured in detecting multi-level regulatory impacts of local and distant SNPs.

To evaluate the expression prediction performance of single-tissue models, Fryett et al.[22] compared prediction models including lasso, ridge regression, elastic net, best linear unbiased predictor, BSLMM, and random forest with eQTL panels of European or African origin. The BSLMM demonstrated the highest prediction accuracy, followed by random forest, elastic net, and lasso. Furthermore, Okoro et al.[23] evaluated the performance of machine learning models, including elastic net,

**Table 1 Reference panel data resource.**

| Dataset | Tissue | samples | Ancestry | Updated year | PMID | Link |
|---|---|---|---|---|---|---|
| GTEx | 54 tissues | 15,201 | White, AA, AS, Others | 2019 | 32913098 | https://gtexportal.org/ |
| TCGA | 67 tissues | 8094 | White, AA, AS, NA, Others | 2021 | - | https://portal.gdc.cancer.gov/ models available in http://gusevlab.org/projects/fusion/ |
| DGN | Whole Blood | 922 | EU | 2014 | 24092820 | https://explorer.nimhgenetics.org/ models available in http://gusevlab.org/projects/fusion/ |
| METSIM | Adipose, Blood | 563 | Fins | 2010 | 28119442 | |
| NTR | Blood | 5339 | Netherlands | 2014 | 20477721 | https://www.ncbi.nlm.nih.gov/gap/?term=phs000486.v1.p1 models available in http://gusevlab.org/projects/fusion/ |
| YFS | Blood | 1264 | Fins | 2011 | 29985430 | |
| CMC | Brain (DLPFC) | 452 | CAU, AA, Hispanic, EA | 2021 | 27668389 | https://www.synapse.org/ |
| eQTLGen Consortium | Blood, PBMC | 31,684 | EU, Others | 2020 | 34475573 | https://www.eqtlgen.org/ |
| GEU | LCL | 462 | CEU, Fins, Others | 2012 | 24037378 | https://www.ebi.ac.uk/biostudies/arrayexpress/studies/E-GEUV-1?query=E-GEUV-1 |
| BrainSeq | Brain (DLPFC) | 175 | CAU | 2015 | 26687217 | http://eqtl.brainseq.org/phase1/ |
| ROS/MAP | Brain (DLPFC) | 2091 | AA, Hispanic, Others | 2020 | 22471860; 22471867 | https://www.synapse.org/ |
| AD Knowledge Portal | Brain, Blood | 2573 | - | 2022 | 33085189 | https://adknowledgeportal.org/ |
| PsychENCODE | Brain | 2198 | - | 2015 | 26605881 | https://psychencode.synapse.org/ |

*DLPFC dorsolateral prefrontal cortex, LCL lymphoblastoid cell line, PBMC peripheral blood mononuclear cell, AA African American, AS Asian, EU European, EA East Asian, FIN Finns, CAU Caucasian, NA Native American, CEU Central European.*

lasso, random forest regression, k-nearest neighbor regression and support vector regression using eQTL data from African, Hispanic and European. The elastic net model showed the highest performance generally, except that the random forest regression model outperformed the elastic net model for some genes. However, none of the current models reach a superior high-level accuracy, so new models still need to be established.

So far, the models mentioned above are built tissue-by-tissue, which means the estimated SNP effect sizes are tissue-specific. However, available sample sizes of many tissues often must be improved to create a powerful prediction model. Moreover, deciding the most suitable tissue to perform TWAS for traits with unclear biological mechanisms is hard. Consequently, prediction models estimating SNPs effects jointly across tissues have been proposed for TWAS analysis. Generally, it has the advantages of taking a larger sample size, which statistically improves prediction power[24], and making full use of similarity in transcriptome-level regulation across different tissues to enhance prediction performance. Hu et al.[25] developed UTMOST, which built a penalized multivariate regression model to predict an expression matrix across tissues, and showed higher predictive accuracy than PrediXcan and FUSION. Further consideration could be given to rank different tissues based on their relevance to traits. By utilizing feature importance measurement, Multi-task Learning Random Forest (MTL-RF)[18] was proposed to evaluate the rank of various tissues (as features) based on their gene expression profile similarity with the putative target tissue. Comparably, Joint-Tissue Imputation (JTI) approach was introduced by Zhou et al.[26] to rank different tissues. It defines the similarity between two tissues based on their DNase I-Hypersensitive Site resemblance, which reflects the genetic regulatory mode. Multiple-tissue-based MTL-RF and JTI performed better predictions than single-tissue-based lasso and random forest models, demonstrating that cross-tissue models may also work better in non-linear models.

Moreover, constructing a prediction model may be challenging due to the limited paired genotype and transcriptome data. Zhang et al.[13] raised SUMMIT, an expression imputation method utilizing integrative eQTL summary data to address the issue. Instead of genotype data, SUMMIT estimates cis-eQTL effect size ($\beta$) with the eQTL summary statistics and a shrinkage estimator of LD reference in a penalized regression framework. The final objective function for optimization can be written as follows:

$$\hat{f}(\beta) = \beta'\hat{R}\beta - 2\beta'\hat{r} + \theta\beta'\beta + J_\lambda(\beta) \qquad (3)$$

In Eq. (3), $\hat{R}$ denotes a shrinkage estimator of the cis-SNPs LD matrix with reference panels such as the 1000 Genomes Project. $\hat{r}$ denotes the estimated eQTL effects (z-scores) from summary-level eQTL datasets. $\theta\beta'\beta$ denotes an L2 penalty term to ensure a unique solution upon optimization. $J_\lambda(\beta)$ denotes the lasso penalty of $\beta$[13]. With large-scale databases of eQTL summary data such as eQTLGen[27], researchers can build prediction models with increased sample sizes. In the paper, the author also compared SUMMIT and conventional TWAS including PrediXcan and FUSION, demonstrating that the larger sample size contributes to better accuracy and TWAS power.

**Models used in the association stage**. Generally, gene-trait associations can be tested by a conventional regression model (e.g., linear, logistic, Cox) or non-parametric model (e.g., Spearman) according to the characteristics of traits in every tissue separately. Differently, kTWAS[28] integrates the principle of kernel-based association test with linear penalty regression to include non-linear SNP effects. In short, it fits SNP weights (same as conventional TWAS) to build linear kernels of a specific genetic region and then conducts sequence kernel association

**Fig. 2 The development timeline of TWAS methods.** Methods with '*' take GWAS summary statistics data as input. Exemplified applications are listed below each method.

analysis. With the linear kernel, kTWAS strengthens its ability to detect gene-trait associations regulated by non-linear genetic effects, thus improving TWAS power.

Likewise, cross-tissue models have different strategies to test gene-trait associations. For example, UTMOST[25] initially performs gene-trait association analysis in each tissue, then unifies the association results through the generalized Berk-Jones test. Differently, MultiXcan[29] builds a multivariate regression model for associating the trait with predicted expressions of a given gene in multiple tissues, and estimates their joint effects using $F$ test.

**Models combining imputation and association stage**. Another strong tendency of TWAS models is to utilize GWAS summary statistics data instead of individual-level data. Individual genotype is challenging to access due to involving private information. Conversely, GWAS summary data is calculated from individual genotypes in a specific population to provide statistics (e.g., effect size) of each SNP on the phenotype, which is more popular for public accessibility. With this situation, Gusev et al.[12] advocated FUSION, a GWAS summary statistics-based model first featured with testing gene-trait association without individual genotype data needed. In FUSION, the expression-trait association is calculated based on trained SNPs weights and GWAS summary data without imputing gene expression. The effect size of a given gene on a trait was defined with the linear combination of the estimated SNP-expression effect sizes and standardized SNP-trait effect sizes, which is subsequently used to analyze the gene-trait associations. Afterward, the potential of utilizing summary statistic data contributes to a surging number of TWAS research without access to GWAS individual data. More summary statistics-based models have been raised based on the form of FUSION. Barbera et al.[11] launched S-PrediXcan using GWAS summary statistic data instead of individual-level data in PrediXcan. Meanwhile, a broad toolset, including PrediXcan, S-PrediXcan, MultiXcan, and S-MultiXcan, was proposed in MetaXcan. Other methods, including UTMOST[25], JTI[26], ETWAS[20] and MOSTWAS[21] also designed their TWAS analysis with GWAS summary statistics data.

Conventional TWAS utilizes ancestry-matched eQTL reference to impute expression and analyze gene-trait associations of the GWAS cohort. However, some TWAS methods have been proposed to utilize eQTL reference from multi-ancestry populations, which maximizes the sample size and improves TWAS power. For example, TESLA offered by Chen et al.[14] was designed

for TWAS, integrating eQTL and GWAS data from diverse ancestries. Briefly, TESLA builds genome-wide allele frequency PCs for different populations and fits phenotypic effects with these PCs by a meta-regression model, which is used to estimate ancestry-matched phenotypic effects. TWAS statistics are then calculated based on the estimated effect and its standard deviation. Moreover, Knutson et al.[15] proposed another multi-ancestry TWAS method, MATS, which can distinguish ethnicity-specific associations. It builds the expression prediction model with the putative SNP effect size ($\beta$) decomposed into three parts: ethnicity-shared effects, ethnicity-specific effects and individual-specific effects. The decomposed $\hat{\beta}$ is separately for association analysis, which enables MATS to identify gene-trait associations at population-shared, population-specific and subject-specific levels.

**Aggregation model**. As the mechanism of SNP regulations varies across many genes, a prediction model with limited SNP effects assumption(s) may be only suitable for specific genetic architectures. Thus, Zeng et al.[30] launched Harmonic Mean $P$ value Aggregated TWAS to aggregate association results from multiple TWAS methods with complementary assumptions. It leverages various $p$ values for the same gene-trait pair from different TWAS models and calculates a final $p$ value using mean harmonic measurement with unaffected false-positive error.

**Other relevant methods**. TWAS has contributed to identify genes with significant associations with traits of interest. However, using relevant and complementary methods jointly to identify causal genes is recommended. Several additional analyses with/after TWAS help to translate association signals into functional or causal units.

For instance, fine-mapping methods prioritize putative causal genes by accounting for LD and pleiotropic SNP effects. FOCUS (Fine mapping Of Causal gene Sets)[31] is commonly applied in post-TWAS analysis, which estimates the probability of a given gene set explaining TWAS signals by a Bayesian framework. Liao et al.[32] determined *FLT3* as a causal gene for Tourette's syndrome with FOCUS. FOGS (Fine-mapping Of Gene Sets)[33] is a later method for fine-mapping, which is featured with a weighted adaptive test method to prioritize causal genes for TWAS results. Zhang et al.[13] identified 11 putative causal genes of COVID-19 with FOGS.

Colocalization is a widely-used approach to test shared genetic basis for GWAS and eQTL to identify target genes with regulatory evidence. For example, Al-Barghouthi et al.[34] identified potential causal genes for human bone mineral density by combing TWAS and colocalization. They implemented TWAS and colocalization based on the same eQTL and GWAS data, and finally found 512 TWAS genes with significant colocalized effects.

Mendelian randomization (MR) analysis is another widely-applied method to detect causal factors of complex traits from GWAS data before the rise of TWAS. Briefly, MR leverages genetic variants as instrumental variants (IVs) and interested intermediate factors as the exposure (e.g., smoking) to infer the causal effects of the exposure on traits of interest as the outcome (e.g., lung cancer). The implementation of MR is under certain strict assumptions: the IVs must be robustly associated with the outcome(s); the IVs affect the outcome only through the exposure; the IVs are independent of the exposure-outcome relationship. Based on these conditions, MR avoids bias induced by unknown confounders and pleiotropy, and can measure the causal effects of the exposure on the outcome. Standard MR calculates the causal effect of one SNP instrument on the outcome with methods such as the ratio of coefficients method, two-stage least square (2SLS) methods, likelihood-based methods, semi-parametric methods and so on[35]. Mathematically, TWAS can be viewed as a two-sample MR analysis with eQTL panel and GWAS panel implementing 2SLS method independently, which aims to infer causal effect from gene expression to the trait[36].

MR focuses on causal inference of the exposure variable, which can be combined with TWAS to identify target genes associated with traits. In particular, Zhu et al.[37] developed summary MR (SMR), which applied 2SLS to test the causal effects of gene expression on interested traits by exploiting gene expression as the exposure and traits as the outcome based on GWAS summary statistics and eQTL data. Yang et al.[38] applied SMR and TWAS to identify significant genes of intraocular pressure (IOP). With eQTL data from GTEx and CAGE, and GWAS summary data of IOP, they conducted SMR and identified 19 and 25 genes respectively. With the same eQTL reference and GWAS summary data, they applied TWAS and identified 12 and 4 overlapped genes with SMR for GTEx and CAGE reference panels respectively.

## TWAS applications in complex traits

Complex traits are affected by multiple genes whose effects are difficult to be measured one by one experimentally. Leveraging data from eQTL studies and GWAS, TWAS has detected important trait-associated genes in numerous polygenic phenotypes. Here, we show some examples of complex diseases and physiological phenotypes to demonstrate research procedures and novel findings of TWAS.

**Complex diseases**. Cancer is one of the most complex diseases with high heterogeneity in multiple tissues. At the transcriptome level of different tissues, TWAS has been applied in detecting gene-trait associations in various pathological processes of cancer, including tumorigenesis, metastasis, and immune escape. For example, Boose et al.[39] utilized S-PrediXcan and FUSION software to implement TWAS in lung cancer. They leveraged a lung eQTL dataset including non-tumor lung tissue of 1038 patients as a reference panel and a GWAS summary statistics of 29,266 lung cancer cases and 56,450 controls of European ancestry. TWAS detected 23 most significant genes enriched in the major histocompatibility complex region, among which *AQP3* and *IREB2* showed novel and strong associations with lung cancer. Guided by TWAS results, the following in vitro assay validated that

expression change of these genes accelerated endogenous DNA damage in lung fibroblasts, implying their tumorigenesis potentials. In addition, Bhattacharya et al.[40] have applied PrediXcan and FUSION to detect ethnicity-specific genes associated with breast cancer in diverse populations. They constructed ethnicity-specific prediction models for African American women and white women individually, and found that the two models showed lower accuracy when predicting gene expression for the other race. In the association analysis, four genes associated with breast cancer-specific survival were only detected in African American women but not in white women, which indicated the ethnicity-specific genetic regulation on expression. Thus, more ethnicity-specific TWAS studies are needed to reveal gene-trait associations in diverse races.

Since brain tissue is highly differentiated, nervous system diseases are usually complex[41], and most current eQTL databases have divided brain tissue into many specific regions. For instance, Alzheimer's disease (AD) is featured with progressive injury of the cerebral hippocampus. In a TWAS study, Liu et al.[7] utilized trained prediction models from PredictDB with 11,688 GTEx samples as a reference. A meta-analysis of AD GWAS with 71,880 patients and 383,378 controls was used as a discovery cohort. They identified 24 AD-associated genes with the S-PrediXcan method, especially the newly reported *PTPN9* and *PCDHA4* expressed in the cerebral hippocampus. Functional annotations and animal experiments showed their functions in synthesizing neurotoxic Aβ peptides and neuronal connections. Thus, further validation research may demonstrate their roles in AD pathology.

Disorders of the immune system usually have general effects on the whole body with poorly understood mechanisms[42], and TWAS helps locate critical genes. Inflammatory bowel disease (IBD) is typically mediated by immune dysfunction. Díez-Obrero et al.[43] generated prediction models with their own eQTL data sets of colon tissues from 445 samples, combining GWAS summary statistics of 60,000 samples from a public resource. By S-PrediXcan, they reported 39 novel colon-specific genes and 19 novel immune cell-specific genes were associated with IBD. For instance, *TRIM31*, *CLDN4*, and *WNT4* with colon-specific associations functioned in epithelium barrier maintenance. Especially, *WNT4* was found to impact colon epithelium fibrosis, providing potential drug targets for anti-fibrosis in IBD.

Along with the essential diseases mentioned above, TWAS has also identified critical genes in lots of complicated conditions, including respiratory disease (asthma[44], idiopathic pulmonary fibrosis[45]), endocrine diseases (diabetes[29]), cardiovascular diseases (acute myocardial infarction[46], coronary artery disease[47]) and so on. Followed by associated genes identified in TWAS, downstream analyses such as finding pathological pathways and gene interaction networks can potentially refine drug discovery and clinical treatments.

**Physiological traits**. Heritable factors impact human traits such as body height, weight, lean length, etc. By S-MultiXcan[11], a large-population analysis was conducted towards multiple traits related to body measurements, utilizing public GWAS summary data from UK Biobank and prediction models from the PredictDB platform. For many traits, each is associated with hundreds of genes even with a strict threshold, implying the complexity of these traits, including height, weight, head circumference, lean mass, base metabolism rate, and body mass index. Efforts of cataloging and explaining the interaction of the enormous volume of genes are expected to demonstrate the genetic regulation beyond body traits.

Besides observed phenotypes, cellular traits also gain much attention in current studies. For example, previous studies have

shown that hematocyte phenotypes are highly heterogeneous in different populations[48]. Wen et al.[49] performed TWAS to pinpoint hematocyte trait-associated genes in the African and Latin populations. This study used two reference panels with 922 European and 610 African/Latin samples to build expression prediction models, and GWAS data were from 10 African/Latin-ancestry cohorts. A linear mixed model was used to test associations between genes and blood cell traits, including hemoglobin, hematocrit, white blood cell count, and platelet counts. Notably, 26 associated genes are found with African/Latin reference panel but could not be identified by European eQTL reference, which indicates some gene-trait associations are specific to certain ethnicities. As a result, the need to establish larger non-European eQTL reference panels is still surging for ethnicity-specific TWAS analysis.

In addition, many biomarkers have attached high attention to disease and biological processes, whose associated genes could be identified by TWAS. For example, abnormalities of lipid proteins and metabolites are observed in many diseases, such as in cardiovascular disease[50]. Andaleon et al.[51] used 44 trained single-tissue models from GTEx (V6), containing at least 70 samples for each, to predict gene expression of GWAS 11,103 samples recorded as Hispanics. PrediXcan-based TWAS identified 14 novel associated genes of lipid-related traits in the Hispanic population, including total cholesterol, triglycerides, low-density lipoprotein, and high-density lipoprotein in blood. Among them, *CCL22* and *ICAM1* were implicated in cardiovascular traits, indicating their association with lipid traits. Imaging helps provide an intuitive observation of brain structures and many neuroimaging traits have been measured as brain phenotypes. With the UTMOST model building on GTEx tissues, Zhao et al.[52] identified 918 significant gene-traits associations between 278 genes and 152 neuroimaging traits using UK Biobank GWAS summary statistics. Among them, 16 genes were found to have significant associations with more than ten traits, which indicated their widespread roles in brain structures.

### TWAS database resources

Available TWAS data is surging and urges a comprehensive database to integrate these TWAS resources. Several TWAS databases have been proposed to store the results of TWAS statistics and provide additional analysis tools with different characteristics.

**TWAS-hub**. Gusev et al. carried out the first TWAS database (TWAS-hub, http://twas-hub.org/) in 2018. It collected raw data from 2010 to 2018 and applied FUSION in its universal TWAS analysis pipeline. Statistics of 75,951 gene-trait associations for 342 traits are provided in the website browser for readers to search and download by trait or gene. Prediction models are built from eQTL data of GTEx and TCGA databases, and genotype data were from UK BioBank and several GWAS summary data sets.

**webTWAS**. WebTWAS (http://www.webtwas.net/) was released in 2021[53]. It is featured in building a collective TWAS framework and providing an online TWAS analysis tool. The framework contains 47 GTEx tissues as reference panels and pre-curated public GWAS summary data from GWAS Catalog, PGC, GWAS ATLAS, etc. The pipeline embedded three popular software, FUSION, UTMOST, and PrediXcan/S-PrediXcan, and TWAS statistics by each model are stored in the database. Updated by 2021, it collects 276,868 gene-trait associations for 1394 fine-mappable GWAS summary statistics. Specifically, webTWAS provides an online tool with 47 GTEx tissues as reference panels

and seven prediction models. Users can handily implement TWAS analysis with uploaded GWAS summary data and parameters on the web server.

**TWAS Atlas**. TWAS Atlas is a knowledgebase that enables a comprehensive collection of TWAS statistics from publications and further data integration with trait ontology and gene-trait knowledge graph[54]. By manual curation from TWAS publications, TWAS Atlas stores high-quality TWAS statistics with 401,266 gene traits towards 257 traits and 135 tissues until 2022. Compared with previous databases based on raw data and uniform analysis pipelines, TWAS Atlas is featured utilizing new-developed methodologies from each publication rather than fixed models and providing research metadata and links. In addition, a trait ontology system was built to unify trait definitions and categories from different studies. Exclusively, based on integrated TWAS results from considerable research, TWAS Atlas constructs a knowledge graph centering on a given trait or gene. The knowledge graph is built on all gene-trait associations from different TWAS and SNP-gene regulatory relationships from GTEx, reaching visualization of multiple and interactive SNP-gene-trait relationships.

### Discussion

With improved models and computational tools, TWAS has been applied in prioritizing gene-trait associations in complex diseases and traits. As a result, this brief review aims to introduce research progress related to TWAS, including its methods from initial to updated models, applications in analyzing complex traits, and available database resources to provide an informative reference for later TWAS researchers. The first section summarized the TWAS framework and introduced different models in each calculation step. We should carefully select models based on SNP regulatory effects on interested genes and trait-related tissues. In the second section, we summarized TWAS applications with examples to demonstrate the data source, analysis design, novel findings in TWAS research, and how TWAS results benefited subsequent functional assays and indicated potential targets in medical research. In the third section, we generalized three TWAS-related databases integrating TWAS available results and laying the foundations for further analysis. We summarized their data sources, processing framework, statistic storage, and unique tools such as online analysis and knowledge graph visualization to offer the most updated view of current TWAS databases.

Specifically, with so many TWAS methods, researchers should fit the framework into their practical analysis pipeline based on the available data and specific research purposes. We provide a more intuitive reference for selecting available methods shown in Fig. 3.

The design begins with a public data type. For GWAS data, the individual genotype can be inputted into individual-based models; otherwise, summary-based models should be considered with summary statistics accessible only. For reference panel data, consider building prediction models if specific matched data (i.e., genotype and expression data for the same individual) is available, or download particular tissue models from public databases such as PredictDB and FUSION. In particular, eQTL summary statistics can be used to build a prediction model with SUMMIT and its potential extensions in the future. To expand the sample sizes of the reference panel, multiple-ancestry methods such as TESLA and MATS can be used to integrate eQTL data from different populations. Next, the choice of tissue dimension depends on the characters of interesting traits. Single-tissue models are recommended for traits with tissue-specific causalities, such as PrediXcan and TIGAR. In contrast, cross-tissue models

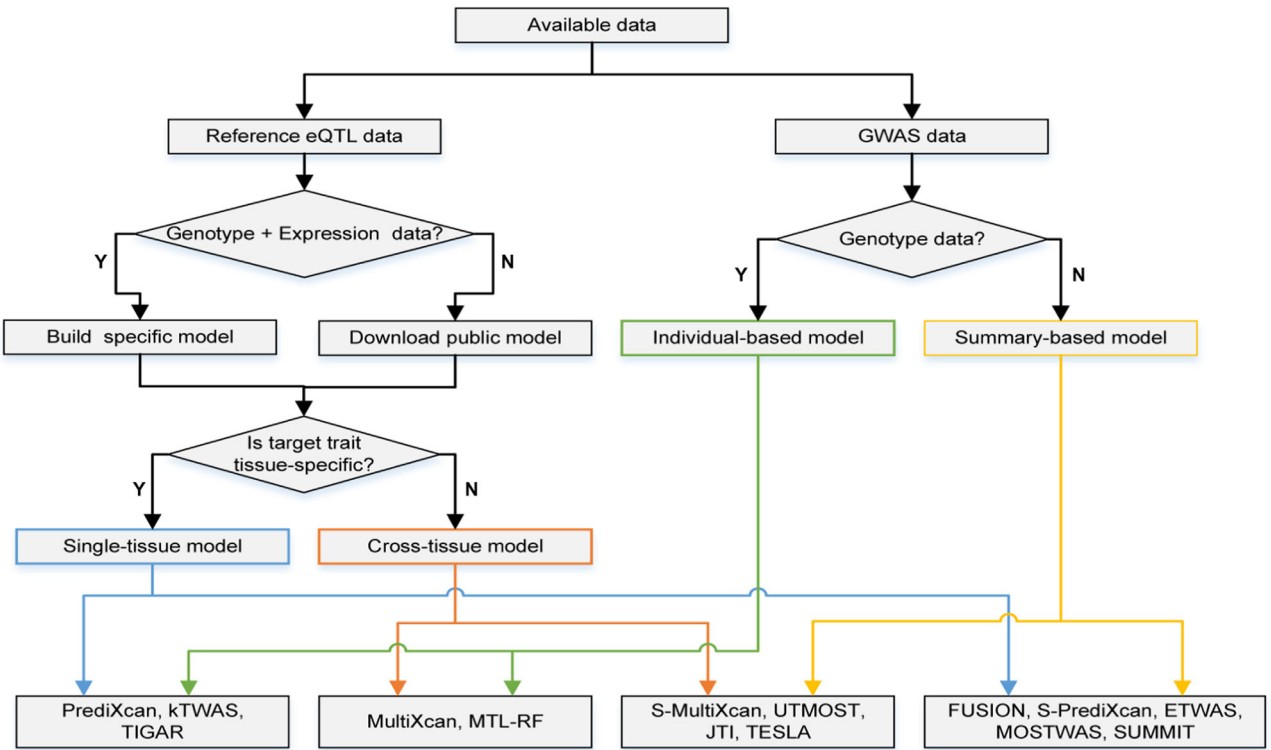

**Fig. 3 Suggestions for TWAS methods selection.** The tree shows the selection process of TWAS models decided by available data and trait characteristics. It involves three main aspects, including the availability of reference panel data, the category of GWAS data, and the preference of tissue type. The joint selection determines the suggested TWAS methods listed at the bottom of the tree.

are more practical in complex diseases involving genetic regulation across multiple tissues. The final choice of the TWAS method is decided by combining individual-based/summary-based models and single-tissue/cross-tissue models. The citation numbers of different methods are summarized in Supplementary Fig. 1b.

Despite the great success in disease-associated gene discovery, TWAS still faces challenges leaving some room for further development. Firstly, the accuracy of expression prediction models is the critical foundation of the following analysis and is expected to be further improved since it is the basis of the subsequent investigation. Current models are mainly based on penalized linear regression (lasso and elastic net), which could not model more complex eQTL effects. Moreover, most existing TWAS studies consider only *cis*-eQTLs when building predictive models. Statistical association testing between *trans*-eQTLs and target gene expression will become computationally expensive and require larger sample sizes to guarantee statistical power. However, aggregated minor *trans*-eQTL effects may significantly impact complex disease-related genes[55]. As investigated, about one-third of variants have *trans*-eQTLs effects mediating gene expression in whole blood tissue[27], suggesting the necessity of considering *trans*-eQTLs in expression. Thus, further models must consider more complex genetic architecture containing different regulatory effects.

Another potential development of TWAS is utilizing single-cell data to detect cell type-specific genes associated with the phenotype. Transcription information mainly comes from bulk RNA-seq data, which covers the expression diversity among different cell types in the same tissue. Single-cell sequencing reveals specific genes expressed in cell subpopulations, especially for highly heterogeneous cell types such as nerve cells, immune cells, tumor cells, etc. Building cell-specific models in

TWAS, it is possible to detect changes in gene expression in different cell types or developing periods, revealing disease pathology and development with a higher resolution. Rising single-cell eQTL data has laid the foundation of single-cell TWAS analysis. Several databases have been constructed, including single-cell eQTLGen Consortium containing blood samples from over 30,000 individuals and GTEx Consortium inferring cell-type specific eQTLs of seven cell types from 35 tissues by computation imputation[56]. Reasonably, single-cell eQTL data helps obtain cell-specific genes associated with diseases and strengthen precise treatments. As single-cell sequencing technology proceeds, more available and accurate eQTL data will be accessible. Thus, combining TWAS and single-cell data could be essential for refiner research.

**Reporting summary**. Further information on research design is available in the Nature Portfolio Reporting Summary linked to this article.

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

## Acknowledgements

This study was supported by Strategic Priority Research Program of the Chinese Academy of Sciences [XDB38030400 to J.X.]; National Natural Science Foundation of China [31970634 and 32170669 to J.X.]; National Key Research Program of China [2020YFA0907001 to J.X.]; the Youth Innovation Promotion Association of the Chinese Academy of Sciences [2022098 to J.Z.].

## Author contributions

J.M. and M.L. produced the figures and wrote the manuscript. Q.G. investigated and collected the documents; J.X. and J.Z. conceived the study and revised the manuscript. All authors read and revised the final manuscript.

## Competing interests

The authors declare no competing interests.
