## [Peer Review File · Communications Biology]

Reviewers' comments:

Reviewer #1 (Remarks to the Author):

The authors reviewed and summarized TWAS methodology, applications, and reference panel data resources, and discussed the challenges and future directions for TWAS studies. It provided a comprehensive overview of TWAS and offered guidance for bioinformatic researchers who are interested in applying TWAS in various circumstances.

Some aspects can be improved as following suggestions:

1. The authors mentioned integrating reference panel with GWAS samples from matched ancestry to avoid disturbing LD patterns among different populations. In a recently published paper by Chen et.al., a novel method, TESLA, which is a multi-ancestry TWAS, is introduced. It would make the reviewing work more comprehensive by reviewing TESLA as well.
2. Some methods can be neatly explained with math equations. It would be helpful to put core math equations together with text for better illustration.

Reviewer #2 (Remarks to the Author):

This is a comprehensive review manuscript on Transcriptome-Wide Association Study (TWAS) by Mai et al. The manuscript is generally clear, but I have several suggestions below:

1. The statement, "The threshold of p-value is commonly set as 0.05/ (total number of predicted genes). To control the false-positive rate introduced by multiple tests, Bonferroni correction or Benjamini-Hochberg correction is often applied to calculate adjusted p-values," requires refinement. The first sentence essentially describes the Bonferroni correction. This might confuse readers who are not familiar with these statistical corrections.
2. The claim, "As expected, both models are more accurate and robust in predicting expression levels than polygenic regression models," is unclear and confusing. I do not understand what does "polygenic regression models" mean here.
3. The explanation provided for BSLMM appears to be inaccurate. I recommend revisiting this section for accuracy.
4. TIGAR [16] is described as using a non-parametric Bayesian model, but the model remains linear. The term "non-parametric" refers to the priors. In other words, TIGAR does not account for non-linear effects of genetic variants. The authors should review all methods accurately.
5. In section 2.1, the authors might want to consider discussing recent methods such as SUMMIT (Nature Communications, 2022) that utilize large eQTL summary data (like eQTLgen) to construct expression prediction models.
6. The statement, "MultiXcan [24] builds multivariate regression models in multiple tissues simultaneously for association tests," appears to be incorrect. MultiXcan does not construct multivariate regression models; it tests the association between a gene across multiple tissues and a trait of interest. Other works, such as InTACT (Genetic Epi), and many others, have been proposed following this concept.
7. The authors may discuss the connection between TWAS and Mendelian randomization.

8. The manuscript should also review other relevant works. For instance, many contributions from Wei Pan's group at the University of Minnesota have been overlooked. After TWAS, additional analyses including fine-mapping (FOCUS and FOGS) and colocalization (such as coloc) are typically conducted. These should also be reviewed. Moreover, the application of TWAS to other biomarkers such as methylation, imaging, and proteins should be discussed. Ethnicity-specific TWAS also merits attention and may be reviewed as well.

Correspondences to the referee's comments

Reviewers' Comments to Author

Referee: 1

Comments for the Author

Reviewer #1 (Remarks to the Author):

"The authors reviewed and summarized TWAS methodology, applications, and reference panel data resources, and discussed the challenges and future directions for TWAS studies. It provided a comprehensive overview of TWAS and offered guidance for bioinformatic researchers who are interested in applying TWAS in various circumstances. Some aspects can be improved as following suggestions:"

Comment 1: *"The authors mentioned integrating reference panel with GWAS samples from matched ancestry to avoid disturbing LD patterns among different populations. In a recently published paper by Chen et.al., a novel method, TESLA, which is a multi-ancestry TWAS, is introduced. It would make the reviewing work more comprehensive by reviewing TESLA as well."*

Response: Thanks for your helpful suggestion. We have added TESLA and MATS in the section of the TWAS framework on Page 5 and Page 12:

"Recently, multi-ancestry TWAS methods integrating eQTL data from multiple ethnics have been developed^{14,15}, enabling cross-ethnic reference with a larger sample size."

"Conventional TWAS utilizes ancestry-matched eQTL reference to impute expression and analyze gene-trait associations of the GWAS cohort. However, some TWAS methods have been proposed to utilize eQTL reference from multi-ancestry populations, which maximizes the sample size and improves TWAS power. For example, TESLA proposed by Chen et al.¹⁴ was designed for TWAS integrating eQTL and GWAS data from diverse ancestries. Briefly, TESLA builds genome-wide allele frequency principal components (PCs) for different populations and fits phenotypic effects with these PCs by a meta-regression model, which is used to estimate ancestry-matched phenotypic effects. TWAS statistics are then calculated based on the estimated effect and its standard deviation. Moreover, Knutson et al. ¹⁵ proposed another multi-ancestry TWAS method, MATS, which is able to distinguish ethnicity-specific associations. It builds the expression prediction model with the putative SNP effect size (β) decomposed into three parts: ethnicity-shared effects, ethnicity-

specific effects and individual-specific effects. The decomposed $\hat{\beta}$ is separately for association analysis, which enables MATS to identify gene-trait associations at population-shared, population-specific and subject-specific levels.”

Comment 2: *Some methods can be neatly explained with math equations. It would be helpful to put core math equations together with text for better illustration.*

Response: Thanks for your helpful suggestion. We have added a math equation to describe the basic linear prediction model on Page 7.

“With a reference panel of n individuals, for a given gene g , the relationship of its expression level and multiple SNP variants can be formulated as:

$$Eg \sim \mu + X\beta + \varepsilon \quad (\text{Equation 1})$$

In Equation 1, Eg is a n -vector which denotes the expression level after correction for confounders (such as age, sex, genotype principal components and so on), X is a $n \times p$ genotype matrix with p -vector SNPs (coded as 0/1/2 or genotype dosages) on the same set of n individuals. β is a p -vector of SNP weights which denotes the corresponding eQTL effects for g . ε denotes the error term and μ denotes the intercept term, which can be dropped after Eg and X are centered at zero.”

We added an equation to demonstrate the cost function of elastic-net regression on Page 8:

“In particular, the elastic net modeling estimates β with a linear combination of lasso (L1) and ridge (L2) penalties by:

$$\hat{\beta} = \underset{\beta}{\operatorname{argmin}} \left[\|Eg - X\beta\|_2^2 + \lambda \left(\alpha \|\beta\|_1 + \frac{1}{2}(1 - \alpha)\|\beta\|_2^2 \right) \right] \quad (\text{Equation 2})$$

In Equation 2, $\|\cdot\|_1$ denotes the L1 norm, $\|\cdot\|_2$ denotes the L2 norm. α denotes the proportion of L1 penalty, which is usually set as 0.5 (as in PrediXcan). λ denotes the penalty parameter, which is estimated by cross-validation⁵.”

We added an equation to describe the differences of SUMMIT with conventional TWAS prediction models on Page 10:

“Moreover, constructing a prediction model may be challenging due to limitation of paired genotype and transcriptome data. To address the issue, Zhang et al.¹³ raised SUMMIT, an expression imputation method utilizing integrative eQTL summary data to address the issue. Instead of genotype data, SUMMIT estimates cis-eQTL effect size (β) with the eQTL summary statistics and a shrinkage estimator of LD reference

in a penalized regression framework.

$$\hat{f}(\beta) = \beta' \hat{R} \beta - 2 \beta' \hat{r} + \theta \beta' \beta + J_{\lambda}(\beta) \quad (\text{Equation 3})$$

In Equation 3, \hat{R} denotes a shrinkage estimator of the cis-SNPs LD matrix with reference panels such as the 1000 Genomes Project. \hat{r} denotes the estimated eQTL effects (z-scores) from summary-level eQTL datasets. $\theta \beta' \beta$ denotes an L2 penalty term to ensure a unique solution upon optimization. $J_{\lambda}(\beta)$ denotes the lasso penalty of β ¹³. With large-scale databases of eQTL summary data such as eQTLGen²⁸, researchers are able to build prediction model with increased sample sizes. In the paper, the author also compared SUMMIT and conventional TWAS including PrediXcan and FUSION, and demonstrated the larger sample size contributes to better accuracy and TWAS power.”

Referee: 2

Comments for the Author

Reviewer #2 (Remarks to the Author):

This is a comprehensive review manuscript on Transcriptome-Wide Association Study (TWAS) by Mai et al. The manuscript is generally clear, but I have several suggestions below

Comment 1: *“The statement, “The threshold of p-value is commonly set as 0.05/ (total number of predicted genes). To control the false-positive rate introduced by multiple tests, Bonferroni correction or Benjamini-Hochberg correction is often applied to calculate adjusted p-values,” requires refinement. The first sentence essentially describes the Bonferroni correction. This might confuse readers who are not familiar with these statistical corrections.”*

Response: Thanks for your helpful comments. We have made a clearer explanation on Page 5 in the revised manuscript.

“(3) Association stage: this step implements hypothesis tests between predicted gene expression and the target trait with different statistic association models. Finally, it pinpoints significant trait-associated genes with their effect sizes calculated. To avoid false positive results induced by multiple testing, statistical corrections are applied to adjust p-values, including Bonferroni correction, Benjamini-Hochberg correction and so on.”

Comment 2: *‘The claim, “As expected, both models are more accurate and robust in*

predicting expression levels than polygenic regression models," is unclear and confusing. I do not understand what does "polygenic regression models" mean here.'

Response: Thanks for your helpful comments. We refer polygenic regression model as a model that fits multiple SNPs and the quantitative target (e.g., cholesterol level, gene expression, disease susceptibility). To clarify the sentence, we have modified the sentence on Page 8 in the revised manuscript.

"The result showed that elastic net and lasso are more accurate and robust in predicting expression levels than simple polygenic models⁵"

Comment 3: *"The explanation provided for BSLMM appears to be inaccurate. I recommend revisiting this section for accuracy."*

Response: Thanks for your helpful comments. The section has been modified on Page 8 in the revised manuscript.

"Another widely-used software, FUSION, adopts the Bayesian sparse linear mixed model (BSLMM)¹², which is a hybrid of the Bayesian variant selection model (a sparse regression model) and a linear mixed model. The BSLMM model was early used in polygenic modeling for polygenic risk scoring and phenotype prediction¹⁶, then is used to estimate the distribution of genetic effect size (β) on given gene locus in FUSION. In TWAS, the two models hold diametrically opposite assumptions about the distribution of β . The linear mixed model assumes that the β is normally distributed, while the Bayesian variant selection model assumes a sparse distribution of β with a small proportion of cis-SNPs having important effects on gene expression. Consequently, the combined model takes advantage of both assumptions and can adapt flexibly to real eQTL effect size distribution when the real genetic mechanism is unknown."

Comment 4: *"TIGAR [16] is described as using a non-parametric Bayesian model, but the model remains linear. The term "non-parametric" refers to the priors. In other words, TIGAR does not account for non-linear effects of genetic variants. The authors should review all methods accurately."*

Response: Thank you for the helpful comments. The section has been modified on Page 9 in the revised manuscript.

"TIGAR²¹ used a non-parametric Dirichlet process regression (DPR) model to capture more genetic effects and achieve more robust performance. The Latent

Dirichlet process introduces an unknown distribution on the variance parameter of SNP effect size (β) and estimates β based on inputted data rather than parametric priors. In the paper, the model comparison showed that the non-parametric model had more substantial TWAS power than PrediXcan²¹.”

Comment 5: “In section 2.1, the authors might want to consider discussing recent methods such as SUMMIT (Nature Communications, 2022) that utilize large eQTL summary data (like eQTLgen) to construct expression prediction models.”

Response: Thanks for your helpful suggestions. The section has been supplemented on Page 4 and Page 10 in the revised manuscript.

“To build an expression prediction model, SUMMIT¹³ was developed to utilize eQTL summary statistics and LD information from reference genomes.”

“Moreover, constructing a prediction model may be challenging due to limitation of paired genotype and transcriptome data. To address the issue, Zhang et al.¹³ raised SUMMIT, an expression imputation method utilizing integrative eQTL summary data to address the issue. Instead of genotype data, SUMMIT estimates cis-eQTL effect size (β) with the eQTL summary statistics and a shrinkage estimator of LD reference in a penalized regression framework.

$$\hat{f}(\beta) = \beta' \hat{R} \beta - 2 \beta' \hat{r} + \theta \beta' \beta + J_{\lambda}(\beta) \quad (\text{Equation 3})$$

In Equation 3, \hat{R} denotes a shrinkage estimator of the cis-SNPs LD matrix with reference panels such as the 1000 Genomes Project. \hat{r} denotes the estimated eQTL effects (z-scores) from summary-level eQTL datasets. $\theta \beta' \beta$ denotes an L2 penalty term to ensure a unique solution upon optimization. $J_{\lambda}(\beta)$ denotes the lasso penalty of β ¹³. With large-scale databases of eQTL summary data such as eQTLGen²⁸, researchers are able to build prediction model with increased sample sizes. In the paper, the author also compared SUMMIT and conventional TWAS including PrediXcan and FUSION, and demonstrated the larger sample size contributes to better accuracy and TWAS power.”

Comment 6: ‘The statement, “MultiXcan [24] builds multivariate regression models in multiple tissues simultaneously for association tests,” appears to be incorrect. MultiXcan does not construct multivariate regression models; it tests the association between a gene across multiple tissues and a trait of interest. Other works, such as InTACT (Genetic Epi), and many others, have been proposed following this concept.’

Response: Thanks for your helpful suggestions. The section has been modified on Page 11 in the revised manuscript.

“Differently, MultiXcan³⁰ builds a multivariate regression model for associating the trait with predicted expressions of a given gene in multiple tissues, and estimates their joint effects using F-test.”

Comment 7: *“The authors may discuss the connection between TWAS and Mendelian randomization.”*

Response: Thanks for your insightful comments. The section has been supplemented on Page 14 in the revised manuscript.

“Mendelian randomization (MR) analysis is another widely-applied method to detect causal factors of complex traits from GWAS data before the raise of TWAS. Briefly, MR leverages genetic variants as instrumental variants (IVs) and interested intermediate factors as the exposure (e.g., smoking) to infer the causal effects of the exposure on traits of interest as the outcome (e.g., lung cancer). The implementation of MR is under certain strict assumptions: the IVs must be robustly associated with the outcome(s); the IVs affect the outcome only through the exposure; the IVs are independent of the exposure-outcome relationship. Based on these conditions, MR avoids bias induced by unknown confounders and pleiotropy, and can measure the causal effects of the exposure on the outcome. Standard MR calculates the causal effect of one SNP instrument on the outcome with methods such as ratio of coefficients method, two-stage least square (2SLS) methods, likelihood-based methods, semiparametric methods and so on³⁶. In particular, Zhu et al.³⁷ developed summary Mendelian randomization (SMR), which applied 2SLS to test the causal effects of gene expression on interested traits by exploiting gene expression as the exposure and traits as the outcome based on GWAS summary statistics and eQTL data.

TWAS and SMR take different methods to calculate the association between genes and traits, which can be combined to detect significant gene-trait associations. Yang et al.³⁸ applied SMR and TWAS to identify significant genes of intraocular pressure (IOP). With eQTL data from GTEx and CAGE as well as GWAS summary data of IOP, they conducted SMR and identified 19 and 25 genes respectively. With the same eQTL reference and GWAS summary data, they applied TWAS and identified 12 and 4 overlapped genes with SMR for GTEx and CAGE reference panels respectively.”

Comment 8: *“The manuscript should also review other relevant works. For instance, many contributions from Wei Pan's group at the University of Minnesota have been overlooked. After TWAS, additional analyses including fine-mapping (FOCUS and FOGS) and colocalization (such as coloc) are typically conducted. These should also be reviewed. Moreover, the application of TWAS to other biomarkers such as methylation, imaging, and proteins should be discussed. Ethnicity-specific TWAS also merits attention and may be reviewed as well.”*

Response: Thanks for your helpful advice. The section has been supplemented on Page 13 in the revised manuscript.

“2.5 Other and relevant methods

TWAS has contributed to identify genes with significant associations with traits of interest. However, using relevant and complementary methods jointly for identification of causal genes is recommended. Several additional analyses with/after TWAS help to translate association signals into functional or causal units. For instance, fine-mapping methods prioritize putative causal genes by accounting for linkage disequilibrium and pleiotropic SNP effects. FOCUS (Fine mapping Of Causal gene Sets)³² is commonly applied in post-TWAS analysis, which estimates the probability of a given gene set explaining the TWAS signals by a Bayesian framework. Liao et al.³³ determined *FLT3* as a causal gene for Tourette's syndrome with FOCUS. FOGS (Fine-mapping Of Gene Sets)³⁴ is a later method for fine-mapping, which is featured with a weighted adaptive test method to prioritize causal genes for TWAS results. Zhang et al.¹³ identified 11 putative causal genes of COVID-19 with FOGS.

Colocalization is a widely-used approach to test shared genetic basis for GWAS and eQTL to identify target genes with regulatory evidence. For example, Al-Barghouthi et al.³⁵ identified potential causal genes for human bone mineral density by combining TWAS and colocalization. They implemented TWAS and colocalization based on the same eQTL and GWAS data, and finally found 512 TWAS genes with significant colocalized effects.”

In the section of listing examples of TWAS applications, we originally considered phenotypes including diverse diseases, observed traits and cellular traits. Inspired by your kind suggestions, we added examples of other traits, such as imaging and metabolite traits

on Page 17. However, we have not observed TWAS study with methylation as biomarker. Association between methylation sites and complex traits is often analyzed with Epigenome-Wide Association Study (EWAS) methods. The combination of TWAS and EWAS contributes to the mechanism insights of traits of interest, for example, Estupiñán-Moreno et al. revealed pathways of giant cell arteritis monocytes in glucocorticoid development by integrating EWAS and TWAS.

“In addition, many biomarkers have attracted high attention in disease and biological processes, whose associated genes could be identified by TWAS. For example, abnormalities of lipid proteins and metabolites are observed biochemical material abnormalities are significant in many diseases, such as lipid disorders in cardiovascular disease⁵⁰. Andaleon et al.⁵¹ used 44 trained single-tissue models from GTEx (V6), containing at least 70 samples for each, to predict gene expression of GWAS 11,103 samples recorded as Hispanics. PrediXcan-based TWAS identified 14 novel associated genes of lipid-related traits in the Hispanic population, including total cholesterol, triglycerides, low-density lipoprotein, and high-density lipoprotein in blood. Among them, *CCL22* and *ICAM1* were implicated in cardiovascular traits, indicating their association with lipid traits. Imaging helps provide an intuitive observation of brain structures and many neuroimaging traits have been measured as brain phenotypes. With the UTMOST model building on GTEx tissues, Zhao et al.⁵² identified 918 significant gene-traits associations between 278 genes and 152 neuroimaging traits using UK Biobank GWAS summary statistics. Among them, 16 genes were found to have significant associations with more than ten traits, which indicated their widespread roles in brain structures.”

Regarding ethnicity-specific TWAS, we have noticed that some TWAS genes are identified for specific ethnic groups. To make the viewpoint clearer, we modified the example on Page 17 as follows:

“Wen et al.⁴⁹ performed TWAS to pinpoint hematocyte trait-associated genes in the African and Latin populations. In this study, two reference panels with 922 European and 610 African/Latin samples were used to predict regulatory weights, and genotype data were from 10 African/Latin-ancestry GWAS cohorts. A linear mixed model was used to test associations between genes and blood cell traits, including hemoglobin, hematocrit, white blood cell count, and platelet counts. Notably, 26 associated genes are found with African/Latin reference panel but could not

identified by European eQTL reference, which indicates some gene-trait associations are specific to certain ethnicities. As a result, the need to establish larger non-European eQTL reference panels is still surging for ethnicity-specific TWAS analysis.”

In addition, we added a study building ethnicity-specific expression prediction models on Page 15:

“In addition, Bhattacharya et al.⁴⁰ have applied PrediXcan and FUSION to detect ethnicity-specific genes associated with breast cancer in diverse populations. They constructed ethnicity-specific prediction models for African American women and white women individually, and found that the two models showed lower accuracy when predicting gene expression for the other race. In the association analysis, four genes associated with breast cancer-specific survival were only detected in African American women but not in white women, which indicated the ethnicity-specific genetic regulation on expression. Thus, more ethnicity-specific TWAS studies are needed to reveal gene-trait associations in diverse races.”

REVIEWERS' COMMENTS:

Reviewer #1 (Remarks to the Author):

The manuscript has been nicely revised. Only several minor comments as follows:

Line 134: suggest to change " \sim " to "=", cause if with error term, then Eg should be equal to the expression on the right hand side.

Line 306: suggest to delete "and".

Line 363: suggest to spell out "MHC" if it is first used in the text.

Reviewer #2 (Remarks to the Author):

The authors have satisfactorily addressed most of my comments. I only have a minor suggestion regarding the connection between Mendelian randomization and TWAS. TWAS can be viewed as a specific case in Mendelian randomization; see the following two papers for details:

1. Zhou, D., Jiang, Y., Zhong, X. et al. "A unified framework for joint-tissue transcriptome-wide association and Mendelian randomization analysis." *Nat Genet* 52, 1239–1246 (2020).
<https://doi.org/10.1038/s41588-020-0706-2>
2. Xue, Haoran, Xiaotong Shen, and Wei Pan. "Causal Inference in Transcriptome-Wide Association Studies with Invalid Instruments and GWAS Summary Data." *Journal of the American Statistical Association* (2023): 1-27.

Correspondences to the referee's comments

Reviewers' Comments to Author

Referee: 1

Comments for the Author

Reviewer #1 (Remarks to the Author):

Comment 1: *"Line 134: suggest to change "~" to "=", cause if with error term, then Eg should be equal to the expression on the right hand side."*

Response: Thanks for your helpful suggestion. We have changed the math equation on Page 4:

"With a reference panel of n individuals, for a given gene g , the relationship of its expression level and corresponding multiple SNP variants can be formulated as:

$$Eg = \mu + X\beta + \varepsilon \quad (\text{Equation 1})$$

In Equation 1, Eg is a n -vector that denotes the expression level after correction for confounders (such as age, sex, genotype principal components and so on), X is a $n \times p$ genotype matrix with p -vector SNPs (coded as 0/1/2 or genotype dosages) on the same set of n individuals. β is a p -vector of SNP weights which denotes the corresponding eQTL effect sizes for g . ε denotes the error term and μ denotes the intercept term, which can be dropped after Eg and X are centered at zero."

Comment 2: *Line 306: suggest to delete "and".*

Response: Thanks for your helpful suggestion. We have deleted "and" on Page 10.

"Other relevant methods

TWAS has contributed to identify genes with significant associations with traits of interest. However, using relevant and complementary methods jointly to identify causal genes is recommended. Several additional analyses with/after TWAS help to translate association signals into functional or causal units."

Comment 3: *Line 363: suggest to spell out "MHC" if it is first used in the text.*

Response: Thanks for your helpful suggestion. We have spelled out "MHC" on Page 12.

"TWAS detected 23 most significant genes enriched in the major histocompatibility complex (MHC) region, among which *AQP3* and *IREB2* showed novel and strong associations with lung cancer."

Referee: 2

Comments for the Author

Reviewer #2 (Remarks to the Author):

Comment 1: *“I only have a minor suggestion regarding the connection between Mendelian randomization and TWAS. TWAS can be viewed as a specific case in Mendelian randomization.”*

Response: Thanks for your helpful comments. We have supplemented the demonstration of the mathematical connection and combined application of MR and TWAS on Page 11.

“Mathematically, TWAS can be viewed as a two-sample MR analysis with eQTL panel and GWAS panel implementing 2SLS method independently, which aims to infer causal effect from gene expression to the trait³⁶.

MR focuses on causal inference of the exposure variable, which can be combined with TWAS to identify target genes associated with traits. In particular, Zhu et al.³⁷ developed summary Mendelian randomization (SMR), which applied 2SLS to test the causal effects of gene expression on interested traits by exploiting gene expression as the exposure and traits as the outcome based on GWAS summary statistics and eQTL data. Yang et al.³⁸ applied SMR and TWAS to identify significant genes of intraocular pressure (IOP). With eQTL data from GTEx and CAGE, and GWAS summary data of IOP, they conducted SMR and identified 19 and 25 genes respectively. With the same eQTL reference and GWAS summary data, they applied TWAS and identified 12 and 4 overlapped genes with SMR for GTEx and CAGE reference panels respectively.”